# Organophosphorus Poisoning: Acute Respiratory Distress Syndrome (ARDS) and Cardiac Failure as Cause of Death in Hospitalized Patients

**DOI:** 10.3390/ijms24076658

**Published:** 2023-04-03

**Authors:** Giuliano Pasquale Ramadori

**Affiliations:** Faculty of Medicine, University of Göttingen, 37073 Gottingen, Germany; gramado@med.uni-goettingen.de or giulianoramadori@gmail.com; Tel.: +49-151-4246-1999

**Keywords:** organophosphate poisoning, dehydration, hypotension, acetylcholine, albumin, acute respiratory distress syndrome, heart failure, atropine, blood volume, fluid replacement, albumin infusion, hypoxic tissue damage, acute-phase reaction

## Abstract

Industrial production of food for animals and humans needs increasing amounts of pesticides, especially of organophosphates, which are now easily available worldwide. More than 3 million cases of acute severe poisoning are estimated to occur worldwide every year, and even more cases remain unreported, while 200,000–350,000 incidentally or intentionally poisoned people die every year. Diagnostic and therapeutic procedures in organophosphate poisoning have, however, remained unchanged. In addition to several neurologic symptoms (miosis, fasciculations), hypersecretion of salivary, bronchial, and sweat glands, vomiting, diarrhea, and loss of urine rapidly induce dehydration, hypovolemia, loss of conscience and respiratory distress. Within hours, signs of acidosis due to systemic hypoxia can be observed at first laboratory investigation after hospitalization. While determination of serum-cholinesterase does not have any diagnostic value, it has been established that hypoalbuminemia alone or accompanied by an increase in creatinine, lactate, or C-reactive protein serum levels has negative prognostic value. Increased serum levels of C-reactive protein are a sign of systemic ischemia. Protective mechanical ventilation should be avoided, if possible. In fact, acute respiratory distress syndrome characterized by congestion and increased weight of the lung, accompanied by heart failure, may become the cause of death. As the excess of acetylcholine at the neuronal level can persist for weeks until enough newly, locally synthesized acetylcholinesterase becomes available (the value of oximes in reducing this time is still under debate), after atropine administration, intravenous albumin and fluid infusion should be the first therapeutic interventions to reestablish normal blood volume and normal tissue oxygenation, avoiding death by cardiac arrest.

## 1. Introduction

The continuous increase in the world’s human, domestic and farm animal populations has dramatically increased the need for (cheap) nutrients for humans and animals. To satisfy this need, the extension of agriculture has been increasing continuously in several areas of the world [1].

Besides the consequences of epidemics due to the transmission of infectious agents from animals to animals and from animals to humans [2,3,4], however, we see a more silent but also devastating epidemic, namely one that is due to acute intentional and non-intentional poisoning with insecticides. In fact, the use of organophosphorus insecticides for widespread occupational and non-occupational purposes has continuously increased over the last six decades [5,6,7], during which it has become more and more intensive.

Pesticides are largely used in agriculture in an attempt to diminish the damage to crops by insects, mites, nematodes, fungi, rodents, birds, and weeds and increase the production of vegetables and fruits, nuts, and wine [6,7]. In addition, pesticides are used not only in agricultural but also in residential, recreational, and other settings [8,9]. The increasing use is not restricted only to so-called third world countries, which 30 years ago accounted for an estimated 25% of the global total in consumed pesticides [10], but it has also further “invaded” the developed world [11].

This causes an annual 200,000–350,000 deaths due to acute poisoning [11,12,13,14,15,16] not only in adults and also children [17,18,19,20,21,22,23,24]. Probably many more cases (several million) of poisoning in human subjects who are “chronically” exposed to pesticides every year remain undetected [25,26,27,28,29,30].

In fact, in 2007, when approximate estimates were available, between 2.1 [10] and 5.1 [11] billion pounds of pesticides were used in the USA. This was considered to be about 22% of the amount of pesticides used worldwide.

Among pesticides, the use of insecticides with different grades of acute toxicity has become very popular. In the USA [29], the use of about 550 million pounds was reported for 2007, also for eliminating the transmission of viral diseases such as west Nile or dengue fever transmitted by mosquitoes [12].

Although rules for the use of these extremely toxic compounds have been issued in all countries of the world, and the WHO continuously reports on toxicity numbers, the numerous publications about acute and “chronic” occupational and non-occupational toxic consequences for children [19,22,23,24] and adults [27,28,29,30,31,32,33,34,35,36] around the world can only be considered anecdotical or the tip of the iceberg.

In the age of globalization and the transfer of foodstuffs from one continent to another, it is, therefore, mandatory to consider that residues of pesticides can contaminate foodstuffs and cause serious harm for consumers everywhere worldwide [26].

In this review, the consequences of reduced degradation of acetylcholine at the neuronal level on the one hand, and dehydration on the other hand, will be discussed. The clinical consequences of dehydration at the systemic level and the mechanisms of the physiological hepatic reaction to direct and indirect stimuli will be described. This will help to interpret laboratory findings and to guide therapeutic interventions.

## 2. Organophosphorus Compounds as Insecticides and as Nerve Agents

Organophosphorus compounds are components of the most efficient insecticides [37,38,39], and organophosphorus poisoning has been reported since their introduction at the beginning of the 1950s, when their mechanism of action was also suggested and then further confirmed [40].

These cases of poisoning occurred more and more frequently, reaching the status of a social calamity [41] or an epidemic.

The organophosphorus compounds bind to a serine residue of acetylcholinesterase in the serum (butyrylcholinetserase, BChE) but also, and more importantly, in the membranes of erythrocytes, (AChE) and in the parasympathetic, sympathetic, and central nervous system, where acetylcholine (ACh) represents the central transmitter that is degraded by the enzyme to interrupt the signal [41,42,43,44,45].

The binding of the poison to the enzyme in the nervous system inhibits the function of the enzyme itself. When acetylcholine is not degraded, the signal transmission continues, leading to overactivation of the single nerve function.

Inhibition is irreversible; therefore, the prolonged availability of ACh in the nervous synapses causes the symptoms of overstimulation, which can only be reduced by administration of an acetylcholine antidote, namely atropine. In fact, there is no drug that can induce a release of the toxic compound from the AChE.

New production of the enzyme in the liver, in the bone marrow and, more importantly, in the synapses must be awaited [46,47,48] before the normal catabolism of ACh in the nervous system is reestablished.

Most cases of poisoning with pesticides/insecticides occur far away from hospital facilities where diagnostic procedures are not readily available. A scale has been introduced to grade the severity based exclusively on the clinical symptoms [40]. The effects of organophosphorus compounds as nerve agents have been studied in human subjects [49,50] and gained heightened attention when they were used in two terroristic attacks in Japan [51,52,53,54,55,56,57], as weapons in Iraq and Syria, in the killing of Kim Jun in Kuala Lumpur, and in assassination attempts in Salisbury and Russia [58].

Based on a prospective study by Grob and Harvey [50,51] of several reports by Japanese scientists who thoroughly studied the victims of the terroristic attacks in Matsumoto in 1994 [52] and in Tokyo in 1995 [53,54,55,56,57], and on numerous retrospective reports from around the world, a great deal of experience has been collected on anamnestic, clinical and laboratory investigations and on therapeutic procedures for dealing with acute and “chronic” [58,59,60,61,62,63,64,65,66] intentional, accidental or occupational exposure to organophosphorus insecticides or nerve agents.

Patients poisoned with cholinesterase inhibitors present the symptoms of acute cholinergic crisis, which causes a toxidrome of muscarinic and nicotinic overstimulation. Patients complain of headache, blurry vision because of miosis, hypersalivation, nausea, vomiting due to gastric hypersecretion, dyspnea and chest tightness due to bronchial hypersecretion, laryngeal constriction and bronchoconstriction, bradycardia, enuresis, uncontrolled defecation, loss of consciousness, sweating, muscle weakness, fasciculation and even flaccid paralysis, all of which are the functional consequences of lack of acetylcholine catabolism at the cholinergic synapses.

Because of patients’ loss of consciousness and of the respiratory symptoms and fear of aspiration, especially when gastric lavage is performed, the POP scale [40,51] is still valid in assessing the severity in organophosphorus intoxication. Therefore, most of the first aid measures concentrate on maintenance of respiratory physiology with appropriate oxygenation, especially in those cases where gastric lavage is performed. This means mechanical ventilation is used as a preventive measure.

The clinically most prominent consequence of the acute cholinergic crisis is hypotension leading to hypovolemic shock, with all of the negative metabolic consequences if it lasts several hours before hospitalization can take place.

The duration of individual hospitalization can be different and causes of death can be directly related not only to the amount of the toxic compound ingested or inhaled, but also to indirect complications sometimes as the consequence of therapeutic interventions.

The mortality rate for acute insecticide poisoning among hospitalized persons both in developed and underdeveloped countries has, however, remained rather high, ranging between 15% and 23%, but in a recent report from India, it has reached 33% [37].

Mortality among mechanically ventilated patients can be above 50% [37].

Although acetylcholine does cause contractile and secretagogue effects without exudative consequences on the nasal and bronchial airways [67,68,69,70,71,72,73,74], the most frequent clinical symptoms are reported to concern the respiratory tract (pulmonary edema), which, despite hypotension, justifies the use of diuretics immediately after hospitalization [70,71]. The most frequent cause of death has been reported to be respiratory insufficiency followed by cardiac arrest, sepsis, hypovolemic shock and others [72,73].

Macroscopic and microscopic autopsy findings in the lungs of those who died within the first 24 h after hospitalization were characterized by interstitial edema and hemorrhage in lungs [70] and tubular degeneration in the kidney (acute tubular necrosis); also, in those who died later after OP poisoning [75,76,77,78,79,80], the findings mostly resemble those observed in people who died of acute respiratory distress syndrome (ARDS) [81,82] of different causes, including viral infections of the upper respiratory tract [83,84,85,86] and also paraquat intoxication [87]. Similar histopathological (microscopic) changes can be induced in minipigs by bronchoscopic instillation of gastric juice containing OP (40% dimethoate 0.5 mL/kg) into the lung [88], simulating aspiration of gastric content, as may happen after ingestion of the insecticide mostly with suicidal intention.

In fact, aspiration increases the duration of hospitalization, morbidity, and the mortality rate, as no therapeutic progress has been achieved in the last 25 years [75].

Other indirect consequences, however, are:

(a) the drastic reduction of intravasal fluid volume (the clinical significance of which has to be emphasized), as consequences such as oliguria and hypotension and shock [73] in the presence of bradycardia may resemble a septic picture and condition the judgment of laboratory findings and therapeutic approach, which also increases the risk of pre-renal kidney failure, as demonstrated by the detection of tubular necrosis at autopsy [76,85] mentioned by Banday et al. [37] and Abdel Baseer et al. [19].

Autopsy investigations have, however, mostly been performed for forensic purposes aiming to demonstrate the cause of poisoning [75,76,80,81].

(b) Under such emergency conditions, because of the direct involvement of the gastrointestinal system (see above) on the one hand and of the lavage of the stomach and or administration of active charcoal to eliminate further absorption of the intoxicant on the other hand, administration of sufficient amounts of proteins and calories is strongly compromised.

The first consequence is the reduction in BChE activity and albumin levels in the serum.

## 3. Acetylcholinesterase and Albumin Measurement: Importance for Diagnosis and Therapy

### 3.1. Acetylcholinesterase

The family of cholinesterases consists of 13 isoenzymes. Eleven of them are synthesized in the liver, distributed in all tissues and called “unspecific” isoenzymes (pseudocholinesterase or butyrylcholinesterase) and released into the serum where their concentration as protein (about 5 mg/L) and their functional activity can be measured [89], and as such, are considered to be markers of liver function [90] and of nutritional status [91]. BChE evolved separately from the acetylcholinesterase (AChE) and is coded by a single gene located on chromosome 3. The plasma protein is a tetramer consisting of four identical chains of 110 kd molecular weight each. Its function has been a controversial matter for decades. It is involved in degradation (hydrolysis) of drugs such as succinylcholine, mivacurium, procaine and tetracaine as well as heroin. However, lack of this enzyme is incompatible with normal life under physiologic conditions [92,93]. Such individuals are at risk of postanesthetic apnea, prolonged coma and eventually death if those compounds have been used [94].

Acetylcholinesterase shares 50% of its sequence with BChE and has a quite similar molecular structure [92].

AChE is expressed in the membrane of blood cells and, most importantly, it is the sole enzyme regulating cholinergic neurotransmission in the brain, skeletal muscle and autonomic nervous system. AChE selectively inactivates acetylcholine within milliseconds after it has been released from presynaptic cholinergic neurons in the brain as well as in the periphery. Both enzymes can be inhibited in the blood and at the synaptic endings by organophosphorus compounds, which then induce a dose-dependent cholinergic overreaction.

The most common method of measurement, however, is to study the functional activity of the enzyme (kU/L). For this purpose, a modified form [95,96] of the colorimetric method of Ellman et al. [97] is used. The enzyme induces the generation of butyrate and of thiocholine from butyrylthiocholine; thiocholine then induces the change of yellow potassium hexacyanoferrate (III) to colorless potassium hexacyanoferrate (II). The colorimeter measures the reduction in the absorption of this dye at 400 nm, which is proportional to the enzyme activity.

In fact, measurement of plasma BChE activity is indicated when ingestion of hepatotoxic nutrients such as toxic mushrooms is suspected, when narcosis is planned, and when symptoms of poisoning with the organophosphorus insecticide parathion (E605) have been ascertained.

The drawback of such a method of measurement in the case of insecticide poisoning is that large inter- and intraindividual differences in AChE [98,99] and especially in BChE [96] activities have been repeatedly reported. A value of BChE activity in the whole blood or in the plasma below the normal range can be interpreted as the consequence of the inhibition of functional activity, while a reduced hepatic synthesis can be responsible for that decrease (see below), as BChE has the shortest half-life, about 24 h [99], of all serum proteins. Therefore, in cases of suspected poisoning with pesticides, to be able to make a final judgment, pre-exposure values would be necessary.

### 3.2. Albumin

#### 3.2.1. Pathophysiology

Human albumin consists of a single polypeptide chain of 585 amino acids with a molecular weight of 66.5 kd. The total amount of albumin in the human body is about 280 g, and the total of serum albumin (normal range, 35–50 g/L) represents about 40% of that (about 120 g). The remaining 60% is contained in the interstitial fluid (about 160 g) at a concentration that is about 60–70% that in the serum [100,101].

From the blood inside the capillaries, albumin flows into the extravascular compartment, returning then to the systemic circulation through the lymphatic vessels. This is accomplished within 16 h [102,103]. There is an exchange of water but not of albumin between the intravasal and extravasal compartments. Therefore, acute loss of intravasal fluid induces a reduction in blood volume and an increase in albumin synthesis in the liver. This can only be efficacious if amino acids are offered through the diet [104,105]. In fact, albumin synthesis is diminished in humans consuming a mainly vegetarian diet [106] and even more after a short fasting [107].

This means that high temperatures or loss of fluids while mountain climbing can increase albumin synthesis in the liver, while urine production is reduced because of fluid loss and hypovolemia [107]. This may be the reason for further albumin and fibrinogen synthesis but only in the upright position, which further aggravates splanchnic hypovolemia [108,109,110].

The skin (about 40%) and the muscle (about 40%) contain most of the interstitial albumin [111]. There is a continuous renewal. Albumin is the most abundant serum protein (50–60% of the total plasma proteins). It is synthesized by the hepatocytes, representing about 40% of the daily protein (12–25 g/d) synthesis [106] by liver cells.

To this end, the hepatocytes need about 800 mg/kg b.w./day of dietary proteins, which represents about 19% of the total recommended dietary intake [94,95]. In fact, fasting time of 18 to 36 h induces an acute reduction in albumin synthesis, of 33% [105,110].

Albumin synthesis is regulated at least in part by changes in oncotic pressure in the liver sinusoid. Albumin structure makes oncotic pressure its most important function to maintain blood volume in continuous equilibrium with the extracellular compartment and, by this, ensuring blood supply for all territories of the body.

In fact, due to its molecular weight, albumin’s contribution to oncotic pressure is about 2.3 times higher than that of the gamma globulins.

Studies in patients with protein-losing enteropathy and those recovering from protein malnutrition [112], or in patients with nephrotic syndrome and major urinary albumin loss [113], show a correlation between the amount of albumin loss and the increase in albumin synthesis in the healthy liver, but not a significant correlation with the calculated oncotic pressure [114], and no correlation with the albumin serum concentration.

Under physiological conditions, albumin represents 80% of the intravascular oncotic pressure.

Every day, about 4% of the body’s albumin is synthesized. This corresponds to a half-life of about 17 days.

The daily synthesis of albumin decreases dramatically if the amount of proteins in an isocaloric diet is reduced. This synthesis also might decrease if the sources of the proteins are vegetables [106].

During constant enteral infusion of a mixed amino acid and glucose solution simulating a sufficient nutritional state, the synthesis of albumin increases by about 20–90%, which results in about 28% of the total protein synthesis in the body, although albumin is only 3% of the total proteins [103], underlining thus the physiological importance of albumin for the vital homeostasis of the body, as the synthesis of fibrinogen remains unchanged. The experiment also showed the importance of insulin for albumin synthesis. In fact, its serum level is also increased when proteins alone are administered with the diet.

The same effect is not observed when amino acid solutions are administered intravenously, suggesting that the liver takes up much more in amino acids from the portal blood immediately after a meal than the amount of amino acids that can reach the liver from the systemic circulation [104]. This is important to be kept in mind when patients who are ventilated in an intensive care unit have to be given nutrients including sufficient amounts of proteins.

While albumin synthesis increases up to the age of 20 years and decreases continuously afterward [115,116], the decrease is more marked in women up to the age of 60 years, when there is no longer any sex difference.

Contrarily, the serum levels of fibrinogen and of ceruloplasmin seem to increase [117,118].

Older people have a lower rate of albumin synthesis than young subjects during fasting [119].

After a liquid meal, the synthesis of albumin as a fraction of the serum albumin, which is renewed every day in younger people (aged 20–35 years), is comparable to that in elderly (>60 y) subjects. The absolute rate of albumin synthesis in aged people is, however, lower than that in the young. This seems to be due to the lower plasma volume in aged persons [120].

There are few doubts concerning the importance of nutrition to maintain a normal albumin plasma level.

This seems to become even more important at an advanced age [119], when especially protein-energy malnutrition may increase [120,121]. Interestingly, in states of chronic reduction of food uptake, albumin plasma levels can be quite normal, underlining the importance of the protein in the vessels compared to that in the interstitial space.

In fact, the necessary amino acids, when not provided by the food, are delivered by the musculature, with a consequent reduction in the body mass index [122].

This must be kept in mind when patients are hospitalized, and low albumin plasma levels are determined [122].

Furthermore, regular alcohol intake may also be responsible for reduced synthesis of albumin and fibrinogen, as ethanol is mainly catabolized in the liver [123].

#### 3.2.2. Albumin, Age, Blood Volume: The Basis for Defense against Homeostasis Disturbances

Acute hypoalbuminemia must be considered separately from long-term hypoalbuminemia when therapeutic options are discussed.

In fact, persons above 65 years of age may show hypoalbuminemia due to a chronic deficiency in their hydration and nutritional status. When replacement therapy for hypoalbuminemia under normal liver function is considered, this must follow a special administration schedule, which has not yet been established.

In critically ill patients such as those poisoned by insecticides or suffering from bacterial or viral infections, eventually albumin replacement strategies must be individually established to maintain normal blood volume, if possible, by first measuring it. Normal blood volume is essential to maintain normal kidney function, which is essential for sufficient urine production without the need for hemodynamic support with vasopressors.

Hypoalbuminemia is a negative prognostic marker [124,125], especially in aged people. This is even more significant when accompanied by an elevated CRP serum level [126] or by increased CRP and IL-6 serum levels [127] in almost all benign [128,129] and malignant pathological conditions that lead to hospitalization [130,131]. It is also a negative prognostic marker for outcome (mortality and morbidity) in patients with cancer under systemic therapy, including liver cancer [132,133,134]. The outcome of patients transferred to the intensive care unit with hypoalbuminemia is poor [128].

Hypoalbuminemia is a negative prognostic marker in surgical interventions for benign or malignant indications [135]. It has been and is still a component of the Child-Pugh score when surgery is indicated in patients with liver cirrhosis [135]. It was also used in decisions regarding indications for liver transplantation until 2001, when the Child-Pugh score was replaced by the MELD score, and serum creatinine replaced serum albumin levels [136].

The advantage of this change, however, could not be confirmed by various authors [137,138,139], even in patients who received transplants with a new liver because of liver cancer [140].

Plasma albumin exerts a kind of buffer function, binding many different compounds such as bilirubin, fatty acids, and divalent cations such as calcium and magnesium, as well as bile acids, zinc, copper, and several drugs. Albumin also acts as antioxidant; in fact, it provides more than 50% of the antioxidant activity of the plasma [141].

Albumin participates in pharmacokinetic and toxicokinetic processes. Of importance is the interaction of albumin with organophosphates, and sites of adduct formation of organophosphates with albumin have been identified [142].

#### 3.2.3. Hypoalbuminemia and Acute-Phase Reaction Markers in Insecticide Intoxication: Analysis of Possible Therapeutic Consequences

There has been a great deal of interest in studying albumin as an efficient hydrolyzer of organophosphate compounds [143,144].

The binding and hydrolysis of the nerve gas soman by human serum albumin was, however, considered to be too slow and of poor relevance for protection against acute poisoning despite the high amount of albumin and the significant amount of organophosphates bound to albumin in the plasma upon in vivo exposure [145]. The detoxification capacity of albumin was found to be relevant when paraoxon, chlorpyrifos oxon, and diazoxon at toxicologically relevant concentrations were used. In fact, incubation of albumin with chlorpyrifos oxon was found to decrease AChE inhibition by 67%, while the inhibition of AChE was much lower when paraoxon or diazoxon were added to albumin (9% and 24%, respectively).

However, the amount of albumin used was significantly lower than the albumin concentration in vivo, suggesting a clinically relevant detoxification role of albumin in vivo, also for the latter compounds [143,144].

Short-term experiments in young volunteers administered the nerve gas sarin showed a reduction in AChE down to zero after oral administration of the poison, but not of albumin, and a relatively fast recovery of BChE activity, beginning at 3 h after administration. Interestingly enough, intravenous administration of plasma or whole blood transiently increased the activity of plasma cholinesterase and of erythrocyte cholinesterase, respectively, but it did not change or prevent symptoms caused by the poison [50].

Decreased albumin concentration in the serum of intoxicated patients has been repeatedly shown to possess negative prognostic significance alone [146] or together with increased CRP serum level [147,148,149,150,151]. In spite of that, routine determination of albumin serum level in intoxicated patients, at hospital admission but also during hospital stay, has not yet been added to the measurement of activity of BChE and AChE as a diagnostic tool. Furthermore, albumin infusion is so far not jet considered a therapeutic option, either in the acute or later phases of the treatment of poisoned patients.

Administration of fresh frozen plasma (FFP) has been used to replace the “consumed” BChE [152], but the positive effect is still controversial [153,154,155,156,157,158].

Some of the positive reports about the use of the administration of different amounts of fresh frozen plasma (FFP) units could eventually be attributed to the positive effect (volume resuscitation) of the total amount of proteins, mostly of albumin contained in the transfused plasma, as has been pointed out by Fulton and colleagues [158]. Some of the side effects are observed also after blood transfusion [159], which could be derived by the “activation” of the tissue macrophages.

This conclusion could be derived from experiments aiming to study the effect of the administration of material known to be taken up by tissue macrophages [160,161,162].

Noh and coworkers recently published a retrospective study analyzing the prognostic value of serum albumin level and the possible mechanisms linking hypoalbuminemia at the time of hospitalization with mortality in patients poisoned with organophosphates [146]. The authors underlined the lack of progress in treating such patients. In fact, the mortality of hospitalized patients ranges between 10% and 20% in all cumulative reports, while the main therapeutic options remain atropine, eventually oximes, and antibiotics. Supportive measures in such patients are mainly restricted to “conservative” fluid administration (without quantitative indications) and eventually diuretics.

Noh and coworkers [146] addressed two questions: (a) can the measurement of serum albumin levels be used to predict the outcome and eventually to develop optimal therapeutic strategies to reduce mortality rates in children and adults with OP poisoning? (b) is the hypoalbuminemia due to malnutrition and inflammation or to the protective effect against OP poisoning?

In the period from 2006 to 2019, patients who were more than 18 years old and admitted to hospital within 24 h after OP ingestion and had serum albumin levels measured (217 patients) were included in the study. Body mass index BMI), type and quantity of OP ingested, time interval between OP ingestion and arrival at the emergency room, vital signs, the available serum level of albumin and BChE activity within 24 h of admission, gastric lavage and the time frame between OP ingestion and gastric lavage, and the time interval between OP ingestion and atropine and eventually PAM (oxime) administration were reported. Development of complications and in-hospital mortality were also studied. The type of OP poison was also classified based on chemical properties. If serum albumin level and BChE activity were measured several times within the first 24 h, delta values were calculated by the following formula: albumin at presentation minus albumin at 24 h divided by albumin at presentation × 100(%). The normal range of the serum albumin level was 35–45 g/L. The serum level of C-reactive protein was measured as a marker of inflammation, and BMI as a marker for nutritional status.

Of the 217 patients analyzed, 40 had a serum albumin level below 35 g/L on presentation; a significant difference compared to the serum albumin levels of those patients who survived (4.1 g/L) was observed.

The patients in the non-surviving group (19.8%) were older and had ingested a larger amount of OP. They had a lower level of arterial bicarbonate and of serum BChE activity. No differences were found between the two groups for OP chemical properties and for the time intervals from ingestion to arrival and from ingestion to atropine and PMA administration. Compared to the normo-albuminemia group, the hypoalbuminemia group had 100% respiratory failure compared to 79% and 70% of the hyperalbuminemia patients. Twenty-five percent had renal failure compared to 6% in the normo-albuminemia vs. 4.5% in the hyperalbuminemia group. Hypotension was significantly more frequently observed in patients with hypoalbuminemia (55%) versus those with normo-albuminemia (34.5%).

The in-hospital cardiac arrest ratio was 32.5% vs. 14.3% vs. 9.1%, hypotension was 55% vs. 34.5% vs 31.8%, and mortality was 45%, vs. 15.8% vs. 9.1% in the hypoalbuminemia, normo-albuminemia, and hyperalbuminemia groups, respectively. 

Interestingly, a significant inverse relationship was found between serum albumin and serum CRP levels.

The authors could not find an explanation for the hypoalbuminemia. One of the possible explanations, however, could be an acute reduction of food intake. Hypotension was most probably due to dehydration and hyponutrition, which together could explain the increase in the serum level of so-called inflammation markers such as CRP because of subclinical tissue hypoxia due to hypovolemia and the consequent tissue hypoperfusion.

Serum CRP and copeptin levels were studied in 100 patients with acute organophosphorus pesticide poisoning (AOPP) who were admitted to the Affiliated Hospital of Taishan Medical University between April 2012 and May 2014 [147] because of the ingestion of pesticides (50–450 mL). Of this group, 54 patients presented symptoms of mild AOPP, 32 of moderate and 14 of severe intoxication. All patients were first treated with gastric lavage, then administered oxygen, atropine and pralidoxime. Intubation and ventilation were performed when necessary. In cases of cardiac arrest, resuscitation and correction of water, electrolyte and nutritional imbalances was performed. Patients with severe AOPP showed increasing serum levels of CRP and copeptin, while the level of both proteins decreased with time in the serum of patients with moderate or mild AOPP, but at day 7 after admission, the serum levels of both proteins continued to be above the norm in all patients. The patients whose serum CRP and copeptin levels were above the median value at day 7 had a worse prognosis.

The serum lactate level was higher in those patients, while BChE activity was lower, and the APACHE II scores were higher. Similar data regarding the significance of the measurement of serum CRP level were found in a study from South Korea [148].

Lee SB [149] and coworkers reported on another prognostic marker in patients with AOPP. The authors found a correlation between the risk of 30-day mortality and a base deficit interpreted as expression of tissue hypoperfusion because of protracted tissue hypotension symptoms found also in cases with septic shock.

In their study, 31 of 154 patients (20.1%) died within 30 days. Those patients showed also a higher leukocyte number, higher BUN, higher serum creatinine concentration, and lower BChE activity, and their APACHE II scores were higher. In the study published by Tang W and coworkers [150], 12 of 71 patients admitted between March 2009 and March 2013 to the emergency department and eventually to the ICU of Fuyang Hospital, Zhejingang Province because of AOPP, died (mortality rate of 16.9%).

All of the patients underwent gastric lavage and diuretic treatment and were given atropine and cholinesterase agents.

Negative prognostic factors on admission were blood lactate, blood pH, base excess, APACHE II score, blood lactate at 6 h postadmission, and duration of hospitalization. The level of BChE in blood plasma and AChE in whole blood were not among the negative prognostic markers and seemed to have only diagnostic value, as they both did not correspond to the quantitative involvement of the cholinergic synapses.

The complex clinical picture is directly induced by overstimulation of the cholinergic nerves and indirectly by the consequences of hypersecretion of the lacrimal, salivary, bronchial, gastric and sweat glands on the one hand and by the additional fluid loss due to vomiting, diarrhea, and uncontrolled loss of urine on the other. In addition, gastric lavage contributes to a delay in the administration of the necessary amounts of calories. These factors contribute to make the causes of the laboratory findings more understandable.

The lack of specific supportive measures turns them (see above) into negative prognostic markers.

In addition, in a recent publication from China [151], mechanical ventilation was a significant risk factor for mortality in patients who were admitted to the emergency room because of chlorpyrifos intoxication.

There was, however, no significant difference between the surviving group and that of non-survivors when respiratory values such as pH, bicarbonate and base excess or BChE activity were considered.

Acute kidney injury was an additional negative prognostic sign [151].

After promising data about the potentially beneficial effect of fresh frozen plasma [152] infused to investigate the increase in the bio scavenger function of fresh frozen plasma/albumin, a preliminary prospective study was performed in the Christian medical college and Hospital in Vallore, India [153].

Patients with organophosphate intoxication were divided into three groups. One group was treated with eight bags of FFP (250 mL each) over 3 days, and the second group was treated with 400 mL 20% albumin over 3 days. The patients in the third group were given 2000 mL of saline over 3 days. Most of the patients were young men (median age 30 years), and the APACHE II scores were comparable, as was the decreased BChE activity. Serum albumin levels were normal and comparable in the three groups. Of the patients in each group, 75% were mechanically ventilated. Eleven of 59 patients—five in the FFP-group, four in the albumin group, and two in the saline group—died.

The treatment performed during the first 3 days after hospitalization did not influence the amounts of free poison compounds in the blood despite the increase in BChE activity, which increased in all groups but significantly in the FFP group [154]. No additional data were collected on the serum albumin and BChE levels in the period during hospitalization up to death.

One patient in the FFP group developed urticaria, and a second patient in the same group developed acute lung injury. Under these conditions, the conclusion was drawn that there was no advantage in administering FFP units or albumin for augmenting their scavenger function.

Similar results were obtained in a second randomized study from Arak, Iran where four packs of FFP were administered to 28 consecutive patients with AOPP and compared with a group of 28 patients who received conventional therapy only. There was no difference in atropine dose administered, length of stay and mortality, even if serum BChE activity was significantly increased by FFP administration [154].

The two negative randomized studies clearly showed that FFP or albumin, as administered in the study of Pichamatu et al. [154], may not have been sufficient to influence the inhibitory effect of OPs at the nerve site even if they could augment the scavenger effects of albumin and BChE.

In a further prospective study from India, however, administration of decreasing amounts of FFPs during the first 3 days after hospitalization was able to reduce ventilation time and mortality compared to the group of patients not receiving additional therapy [155].

Amend N and coworkers [156] administered 10 units of FFP within the first 4 days to an intoxicated patient who presented with hypotension, reduced kidney function, lactic acidosis and almost complete inhibition of serum BChE and had to be mechanically ventilated as a preventive measure. Atropine had to be administered until the clinical signs, miosis and salivation had disappeared (day 15). The neurological effects of Ach were not influenced by the transfused FFP units.

As suggested by Fulton J et al. [158] in their comment on the report of Guven M et al. [152], who found a positive effect of FFP transfusion in OP-intoxicated patients, the use of FFP might be a form of volume resuscitation.

A positive effect on the outcome of OP-intoxicated patients was found in a prospective study from China in a group treated with fresh packed red blood transfusions. The transfusions also reduced the amount of atropine needed [159]. The same effect was not achieved when stored red blood cells were used. The patients, however, showed a level of AChE activity that was quite high on admission (about 1500 KU/L). However, there was no report about the neurological symptoms during their stay in the hospital.

On the other hand, some of the side effects attributed to the administration of fresh frozen plasma or even whole blood units could have derived from a kind of overwhelming effect of the reticuloendothelial system mainly located in the liver but also in the spleen and in the lung, which can have even systemic consequences, as shown in experimental settings [161,162].

Early endotracheal intubation and mechanical ventilation, also as a preventive measure when gastric lavage is necessary, are critical in the treatment of patients with clinical signs of intoxication with organophosphate poisons.

In most of the cases, however, ventilation is generally protracted even when no signs of pulmonary involvement are present.

This means that prolongation of deep sedation and the difficulty of administering the required number of calories via the nasogastric tube, in most of the cases, is accompanied by the administration of antibiotics often because of the persistence of an elevated serum level of CRP, which is interpreted as a sign of bacterial infection. Treatment may become even more complicated when dialysis is performed upon unclear indication [163], sometimes possibly due to administration of diuretics [71,150] and “conservative” fluid policy in patients with low blood pressure.

Respiratory failure (but also cardiac arrest) is one of the main causes of death in such patients [72,73,74,75], several other predictors being cofactors such as age, severity of poisoning and duration of mechanical ventilation.

Autopsies are seldom performed in patients who died because of AOPP, and in most cases, it is related to the search for the toxic agent [77,78,79,80,81,82] and less to mechanisms that lead to death such as cardiac arrest, especially in persons of young age.

If one takes into consideration the assumption that the laboratory changes observed in AOPP patients are mainly the indirect consequence of cholinergic overactivity, the controversial reports about the positive effect of the administration of fresh frozen plasma or of whole blood with the intention of replacing the BChE and AChE activity should be reinterpreted.

The corresponding careful therapeutic corrections of the various imbalances are mandatory and may condition the outcome more than neurological symptoms, as can be learned from the original study by Grob at al. [50] and from the well-documented case reports by Amend [156] and Stindl et al. [164].

The proper interpretation of the origin and significance of markers of tissue damage such as increased levels of C-reactive protein, fibrinogen, lactate, amylase [165] or serum troponin level and other markers of cardiac damage [166] in combination with hypoalbuminemia is of crucial importance in guiding treatment of the “secondary” effects of acute AOPP (Figure 1) if we wish to improve survival in such patients [167,168,169,170] after decades of therapeutic stagnation [171,172,173].

## 4. Conclusions

The leading maxim in the treatment of AOPP intoxication should, therefore, be “treat the patient, not the poison” [13], as has been successfully practiced in recently published case reports [164,167,168,169,170]. Elevated serum levels of inflammation markers in combination with hypoalbuminemia must be interpreted more as signs of tissue ischemia caused by hypovolemia than as direct consequences of the reduced acetylcholine degradation at the neuronal level. Blood transfusions may cause overloading of the liver macrophages with additional hepatic inflammatory reaction. An especially successful case was the infusion of 125 g of albumin together with 6.5 L of fluids and sodium bicarbonate in a patient with severe acidosis when her serum albumin level decreased to 1 g/dL [169], probably due to prolonged hypovolemia after ingestion of a large volume (200 mL) of dichlorvos (DDVP). Addition of oximes does not seem to be helpful in reducing the AChE inhibitory effect of the organophosphate [172,173].

## Figures and Tables

**Figure 1 ijms-24-06658-f001:**
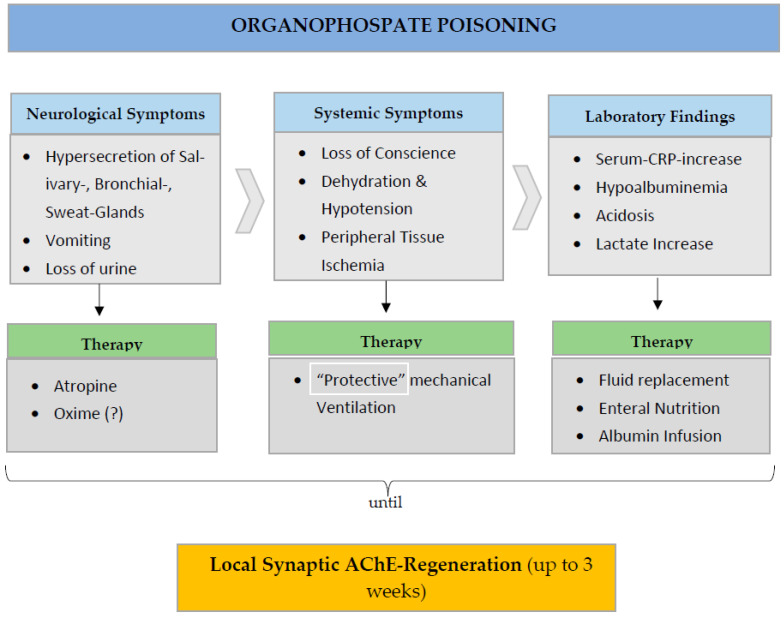
Main clinical and laboratory findings and therapeutic consequences in hospitalized patients after organophosphate poisoning. Production of sufficient “new” AChE at the level of the neuronal synapses can take up to 3 weeks.

## Data Availability

Not applicable.

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
