# Peer review of "Organophosphorus Poisoning: Acute Respiratory Distress Syndrome (ARDS) and Cardiac Failure as Cause of Death in Hospitalized Patients"

_ijms, 2023, doi:10.3390/ijms24076658_

Round 1

Reviewer 1 Report (Previous Reviewer 1)

I appreciated the changes made to the title and the removal of the figures.

The writing style remained quite sloppy. In the first paragraph of the Introduction section, something is missing. Across the manuscript, double spaces between words can be found, or no spaces at all where they are needed. The references in the text are placed between () instead of [].

References between 31-36 are not present in the text.

The abbreviation for acetylcholine (ACh) need correction in some places (e.q. line 94)

The references must be written according to the instructions for authors.

The section about albumine (lines 217-298) could be reduced. 

 The abbreviation FFP is not explained in the text. 

Line 367 - FFF administration

Author Response

The  reviwers´ criticisms were justified and their suggestions were very constructive.

The following changes have been made as a response to

reviwer 1:

  1. The first sentence of the introduction is now complete
  2. The different spaces have been corrected
  3. Proper brackets [] have now been used for the references
  4. The references 31-36 have now been introduced into the text (line 71)
  5. The abbreviation Ach has been corrected
  6. The references are now written according to the instructions for authors
  7. The section about albumine has been reshaped by changing the headlines.
  8. Explanation for the FFP abbreviation has been introduced (line 372).
  9. Please prvide me with  editorial assistance concerning the changes necessary for the references.

Reviewer 2 Report (New Reviewer)

The information provided in the review was inclusive of the overall objectives given the title. The abstract was hard to follow in general. Within the document the sections seemed disjointed with sections not flowing.  I suggest the review be reorganized and the formatting cleaned up to see if overall the document reads well providing a clear conclusion to support the abstract summary.

Author Response

The  reviwers´ criticisms were justified and their suggestions were very constructive.

The following changes have been made as a response to:

Reviewer 2:

To improve the readebility of the manuscript, subheadings have been introduced in the albumine section and further changes have been also introduced on page 8.Furthermore the graphical abstract

has been introduced as Figure on page 12.

This manuscript is a resubmission of an earlier submission. The following is a list of the peer review reports and author responses from that submission.

Round 1

Reviewer 1 Report

The subject is interesting because organophosphate compounds are involved in severe poisonings, but there are language mistakes and many typing and editing errors in the manuscript.

I don’t think the title reflects well the content of the manuscript. The organization of the manuscript should be improved, the information needs to be compiled, and the purpose of this literature review is not clear. The amount of information is high, but not everything is directly and closely related to the subject (See points 4, 5, and 6). The author should select and emphasize only the aspects related to the topic (organophosphate poisoning) for a clearer perspective for the reader.

The quality of the images is poor in most cases, Figures 4 and 5 still have the original captions, but my main objection is that the liver damage was not induced by organophosphate compounds in the chosen figures.

I also have other observations:

The abstract section is far too long, and the author should state the aim of the paper.

Introduction - There is an increasing risk of epidemics in animal farms and of the transmission of infectious agents from animals to humans ,which can cause millions of deaths as it is actually the case for COVID-19 (1).In many other cases of epidemic outbreaks the infectious agent(s) originate from animals(2,3). This is due to the fact, that the number of animals needed for human nutrition increases continuosly. – In my opinion, if the paragraph remains in the current form, there is only a tangential connection with the subject of the paper. I suggest rephrasing or removing it.

Organophosphourus compounds as insecticides and as nerve agents – The author states: symptoms of overstimulation which can only be reduced by administration of an acethylcholine antidot,namely atropine. In fact there is no drug which is able to induce a release of the toxic compound from the AChE. What about oximes (e.g. obidoxime)?

·       Abbreviation – Buthyrylcholinetserase – ? BachE

 Conclusions – are not appropriate for the chosen title.

 Overall, the manuscript needs to go through major revision before being considered for publication.

Author Response

Point by Point answer to the criticisms of the Reviewers:

Reviewer 1.

I am extremly thankful to the reviewer for the very helpful remarks and suggestions and I hope that the following changes I have made are satisfactory .

  1. The title has now been changed
  2. The abstract has been shortened (250 words) and reshaped
  3. The first sentence of the introduction has now been rephrased
  4. A few sentences have now been added to the end of the introduction (lines 81-84).
  5. The original legends of figure 4 and 5 have been deleted.
  6. The reviewer is right.The chosen figures only want to show that what happens in the poisoned persons is not due to the direct interaction of the organophosphates with the liver cells, but is  the reaction of the liver (and also of other organs) to stimuli triggered by the „general“ consequences (tissue ischemia due to hypovolemia) of the poison,which are similar to other noxae e.g viruses)
  7. Three sentences have been added to the conclusion.
  8. The last sentence of the conclusion directly addresses the

question of  therapeutic utility of the administration of the

oximes.

The language problems have been(hopfeully) corrected.

Reviewer 2 Report

I have reviewed a paper: "PANDEMIC" OF ORGANOPHOSPHORUS POISONING:Acute Respiratory Disstress Syndrome (ARDS) and CARDAC FAILURE AS CAUSE OF DEATH.

The paper is interesting, informative and by my opinion, a valuable contribution in knowledge about the processes induced in organism after poisoning has occurred. After some minor corrections, I think this paper can be accepted for publication.

General comment: there are some typographical errors through the text, so please check it and correct it.

Figures 3 and 4 are of bad quality. Please, revise it.

Included figures are results of other Authors, so permissions of them should be included.

Author Response

Reviewer 2:

Thank you very much for the positive judgement.

I hope that all the typographical errors have been corrected.

The figures are autoradiographic  results of the hybridisation of northern blots with radioactive cDNAs.I have deleted the original legend from figure 4.

Permission has been obtained from the publishers as indicated in each legend.

Reviewer 3 Report

This review article entitled “PANDEMIC“ OF ORGANOPHOSPHORUS POISONING: ACUTE RESPIRATORY DISTRESS SYNDROME (ARDS) AND CARDIAC FAILURE AS CAUSE OF DEATH " aimed to discuss an interesting topic. However, it is poorly written and the content does not reflect the main topic. Thus, this review does not meet the standards for publishing in IJMS.

Author Response

Reviewer 3.

I hope that the modified manuscript is more acceptable for this reviewer.

Round 2

Reviewer 1 Report

The author addressed most of my observations.

However, there are still some typing errors (e.g., missing/extra spaces, capital letters) that need to be corrected. Figure 3 is quite blurry.

In his reply, the author explained why he used examples of liver damage not induced by organophosphate compounds -"The chosen figures only want to show that what happens in the poisoned persons is not due to the direct interaction of the organophosphates with the liver cells, but is the reaction of the liver (and also of other organs) to stimuli triggered by the „general“ consequences (tissue ischemia due to hypovolemia) of the poison, which are similar to other noxae e.g viruses)." In my opinion, this text fragment could also be used in the manuscript for a clearer perspective for the reader.

Author Response

I thank again the reviewer for the kind suggestion.I added the sentence to the text (lines 475-478).

I also checked again the manuscript for typos and corrected them to the best of my power.
